# Synthetic CpG islands reveal DNA sequence determinants of chromatin structure

Elisabeth Wachter[1], Timo Quante[1], Cara Merusi[1], Aleksandra Arczewska[1], Francis Stewart[2], Shaun Webb[1], Adrian Bird[1]*

[1]The Wellcome Trust Centre for Cell Biology, University of Edinburgh, Edinburgh, United Kingdom; [2]Genomics and Biotechnology Centre, Technische Universitaet Dresden, Dresden, Germany

**Abstract** The mammalian genome is punctuated by CpG islands (CGIs), which differ sharply from the bulk genome by being rich in G + C and the dinucleotide CpG. CGIs often include transcription initiation sites and display 'active' histone marks, notably histone H3 lysine 4 methylation. In embryonic stem cells (ESCs) some CGIs adopt a 'bivalent' chromatin state bearing simultaneous 'active' and 'inactive' chromatin marks. To determine whether CGI chromatin is developmentally programmed at specific genes or is imposed by shared features of CGI DNA, we integrated artificial CGI-like DNA sequences into the ESC genome. We found that bivalency is the default chromatin structure for CpG-rich, G + C-rich DNA. A high CpG density alone is not sufficient for this effect, as A + T-rich sequence settings invariably provoke de novo DNA methylation leading to loss of CGI signature chromatin. We conclude that both CpG-richness and G + C-richness are required for induction of signature chromatin structures at CGIs.

*For correspondence: A.Bird@ed.ac.uk

**Competing interests:** The authors declare that no competing interests exist.

**Reviewing editor**: Anne C Ferguson-Smith, University of Cambridge, United Kingdom

## Introduction

CpG islands (CGIs) are stretches of atypical genomic DNA sequences that mark most gene promoters (**Bird, 1986**; **Deaton and Bird, 2011**). Though individually unique in nucleotide sequence, mammalian CGIs share two features that often correlate but can vary independently: a G + C-rich base composition (65% vs 40% for the bulk genome) and high density of the dinucleotide CpG (5–10-fold higher than the bulk genome). Whereas bulk genomic DNA is globally methylated, CGIs associated with promoters invariably lack DNA methylation. It has been proposed that CGIs function as generic platforms for gene regulation, probably via proteins that bind to CpG and influence chromatin modification (**Blackledge and Klose, 2011**; **Deaton and Bird, 2011**). This hypothesis has gained credence due to accumulating evidence that enrichment or depletion of specific chromatin marks at CGIs is linked to proteins that bind to unmethylated CpG. So far, all such proteins possess CXXC zinc finger domains that bind specifically to the CpG dyad in duplex DNA (**Lee et al., 2001**). Cfp1, for example, is a CXXC domain-containing component of the Set1/COMPASS complex (**Lee and Skalnik, 2005**), which generates H3K4 methylation, a signature chromatin mark at non-methylated CGIs (**Bernstein et al., 2006**; **Guenther et al., 2007**). Accordingly, Cfp1 is concentrated at CGIs as determined by chromatin immunoprecipitation (ChIP) and its absence is associated with reduced H3K4 methylation at many CGIs (**Thomson et al., 2010**; **Clouaire et al., 2012**). The H3K4 methyltransferases Mll1 and Mll2 are also CXXC proteins and each is found at CGIs in mouse embryonic stem cells (ESCs) as determined by ChIP-Seq (**Hu et al., 2013**; **Denissov et al., 2014**). Similarly the CXXC domain-containing proteins Kdm2a and Kdm2b are enriched at CGIs. Kdm2a is an H3K36 demethylase that contributes to depletion of H3K36me at CGI promoters (**Blackledge et al., 2010**),

**eLife digest** The building blocks of DNA are four molecules commonly named 'A', 'T', 'C' and 'G'. The order of these DNA letters in a gene contains the instructions to make specific proteins or other molecules. Other stretches of DNA contain codes that direct the cell's machinery to genes that need to be switched on or switched off. The start of a gene, for example, has a stretch of DNA called a promoter, which is where the molecular machinery that switches on the gene is assembled.

A human cell can contain over two and half metres of DNA. To get this length to fit inside the cell, the DNA is wrapped tightly around proteins to form a structure called chromatin. However, this packing can make it difficult to access the right gene at the right time. As such, chromatin is often marked with small chemical tags that earmark which genes should be either activated or inactivated, and/or that cause the DNA to unpack.

Most gene promoters contain a sequence of DNA with many Cs and Gs found one after the other, called a CpG island. Researchers have previously shown that the chromatin of CpG islands has two types of chemical markings—one that normally marks active genes, and another that often marks inactive genes. It was suggested that having both kinds of markings allows CpG islands to prime nearby genes, so that they are ready to be quickly switched on or off as the cell develops. However, the features of the DNA sequence in these CpG islands that are important for this process had not been directly tested.

Wachter et al. have now inserted an artificial DNA sequence that included a CpG island into mouse stem cells. The chromatin around these CpG islands was readily marked with both activating and inactivating chemical marks. Furthermore, by changing the sequence of the artificial DNA, Wachter et al. revealed that these chemical marks were only added when the DNA sequences contained a lot of Cs followed by Gs. Other artificial sequences with lots of Cs and Gs, but where Gs were rarely found immediately after the Cs, had neither of the two chemical marks on the chromatin. This suggests that nearby genes would be harder to locate and activate as the cell grows and develops. On the other hand, when the DNA contained a lot of As and Ts, the chemical marks were added directly to the DNA (rather than to the chromatin)—and this prevented both the activating and the inactivating chemical marks being added to the chromatin.

Now that the common features of CpG islands that influence chromatin are known, the next step is to find out how this is achieved. Further work will be needed to uncover which proteins in a cell interpret these DNA sequence such that nearby genes can be switched on or off.

whereas Kdm2b facilitates recruitment of the PRC1 complex to transcriptionally silent CGIs (*Farcas et al., 2012*; *Wu et al., 2013*).

Previous studies have shown that CGI-like DNA sequences can impose an altered chromatin state in ESCs. Promoter-less CGI-like DNA sequences of invertebrate origin in mouse ES cells were initially shown to cause local enrichment of trimethylation of lysine 4 of histone H3 (H3K4me3) and to recruit the CpG binding protein Cfp1 (*Thomson et al., 2010*). Similar experiments using G + C-rich DNA derived from bacteria created chromatin marked by both H3K27me3 and H3K4me3 (*Mendenhall et al., 2010*). Whereas H3K4me3 is characteristic of active promoters, H3K27me3 is associated with transcriptional repression via the polycomb complex. The coincidence of these two marks in so-called 'bivalent' chromatin is thought to be a feature of genes that are poised to become either active or silent during early development (*Azuara et al., 2006*; *Bernstein et al., 2006*; *Voigt et al., 2013*). Many native CGIs in ESCs adopt a 'bivalent' chromatin structure when transcriptionally silent.

Despite evidence that CXXC proteins play a role at CGIs, the hypothesis that CpG density is the critical determinant of CGI function has not been directly tested. Here we vary the DNA sequence composition of artificial promoter-less CGIs to assess the relative importance of G + C-richness and high CpG density. Our assay relies upon chromosomal integration of artificial CGI-like sequences into the genome of ESCs. We show that a bivalent chromatin configuration and absence of DNA methylation represent the default state of biologically inert CGI-like DNA sequences. While a high CpG density is essential, it is not sufficient to guarantee the bivalent chromatin structure, as CpG-rich DNA sequences with a G + C-poor average base composition do not acquire this chromatin signature. In fact A + T-rich, G + C poor insertions with a CpG frequency matching that of CGIs consistently become

DNA methylated in ESCs, although the bivalent configuration can be restored if the dominant DNA methylation is removed. Our findings demonstrate that CpG-richness is essential for the formation of bivalent chromatin, whereas G + C-richness is required to exclude DNA methylation, which when present is dominant over the other chromatin marks.

## Results

### Formation of a novel bivalent domain at artificial CGI-like sequences

Comparison between CGIs and an equivalent number of sequences from the bulk genome shows that both CpG frequency and G + C richness are distinct in CGIs compared with bulk genomic DNA of mouse (*Figure 1A*). A previous study showed that bacterial DNA with CGI-like features organised bivalent chromatin in ESCs (*Mendenhall et al., 2010*). In order to verify and extend this result we introduced CGI-like DNA sequences of ~1000 nucleotide pairs (the average length of native CGIs) into a bacterial artificial chromosome (BAC) containing a human 'gene desert' using recombineering (See *Figure 1—figure supplement 1*). A computer-generated CGI-like sequence (Artificial CGI 1) was designed with a CpG frequency of one per ~10 base pairs and a base composition of 65% G + C (*Figure 1A*, *Figure 1—figure supplement 2* and *Figure 1—source data 1*). A second CGI-like sequence (PuroGFP; *Figure 1A*, *Figure 1—figure supplement 2* and *Figure 1—source data 1*) was of prokaryotic origin, being derived from a promoter-less bacterial puromycin gene adjacent to codon-optimised green fluorescent protein coding sequence (*Thomson et al., 2010*). All constructs lack a promoter, allowing us to focus on the interaction between DNA sequence and chromatin modification without the complicating involvement of transcription. The gene desert regions flanking CGI-like sequences are intended to insulate against effects of the genomic and chromatin environment at different BAC integrations sites.

Three independently transfected stable ESC lines with low copy number random integrations of the BAC containing Artificial CGI 1 and one cell line with the PuroGFP CGI were selected for ChIP analysis (See *Figure 1—figure supplement 1*). As controls we monitored active genes (*Sox2* and *Gapdh*), which are marked by H3K4me3, an endogenous bivalent gene (*Hoxc8*), which carries both H3K27me3 and H3K4me3, and an inter-genic region of chromosome 15 (m15), which bears neither mark. The CGI-like insertions consistently generated a bivalent chromatin structure marked by H3K4me3 and H3K27me3 (*Figure 1B,C* and *Figure 1—figure supplement 1*). We refer to DNA sequence domains as bivalent using the convention that H3K4me3 and H3K27me3 marks coincide at a single integration site. It is possible that some cells in the population harbour an integrant marked by only one of these marks whereas other cells possess only the other mark. This configuration has not been detected at the few ESC bivalent domains tested so far. More likely is that the two marks are interspersed at a given bivalent sequence domain, though we did not test this experimentally (See *Voigt et al., 2013*). The levels of H3K4me3 at the artificial CGI were similar to those at the endogenous bivalent gene *Hoxc8*, whereas H3K4me3 levels in the DNA sequences flanking the CGI were unaffected by the insertion. H3K27me3 was less discrete, spreading variably into flanking regions of the human BAC, but Suz12, a component of the PRC2 complex that deposits the H3K27me3 mark, was tightly localised to the CGI-like sequence in each case. We tested an unrelated artificial CGI-like sequence with similar overall sequence properties (Artificial CGI 2; *Figure 1—figure supplement 1*), this time integrated at a recombination cassette within the beta globin locus of mouse ESCs (*Lienert et al., 2011*). Independent replicate stable transformant cell lines again showed consistent presence of a bivalent chromatin structure (data not shown). The DNA methylation status of the integrated artificial CGIs was investigated by bisulfite sequencing. This showed that all the insertions reproducibly maintained a low level of DNA methylation (*Figure 1D*; *Figure 1—figure supplement 1*). DNA methylation levels at CGI-like sequences inserted into the gene desert (~10%) were consistently somewhat higher than in the cassette exchange system (0.5%) and were somewhat higher than an endogenous control CGI in the same DNA samples (*Dlx5*; ~4–5%, data not shown). Despite this variability, it is evident that all of these artificial DNA sequences maintain a largely non-methylated status.

### H3K4me3 at an artificial CGI is independent of Cfp1 and RNA polymerase II

To test whether these promoter-less artificial DNA sequences were transcriptionally inert, we performed ChIP with antibodies recognising three differentially phosphorylated forms of RNA polymerase II in

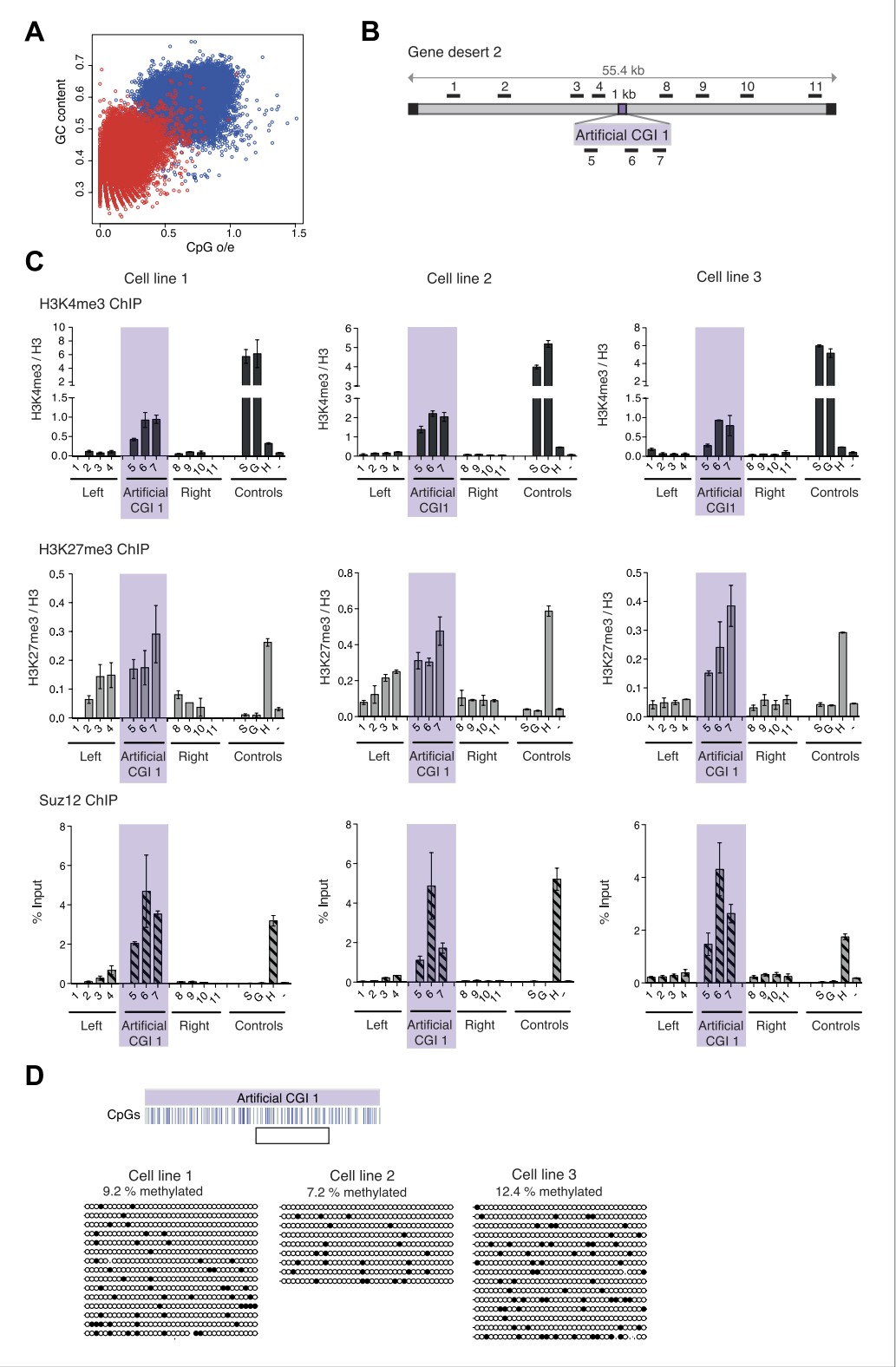

**Figure 1**. A novel bivalent chromatin domain is formed at promoter-less artificial CGI-like sequences integrated within a gene desert in mouse ESCs. (**A**) CpG frequency and G + C content of CGIs in the mouse genome (blue circles) and an equivalent number of equal-sized (1000 base pair) random fragments of bulk genomic DNA (red circles). (**B**) Map of human gene desert 2 (grey bars; Chr1:81,106,616-81,153,886) showing the integration site of the

*Figure 1. Continued on next page*

*Figure 1. Continued*

Artificial CGI-like construct (purple box). Black boxes at the ends indicate bacterial BAC sequences. Black bars above indicate the position of Q-PCR amplicons (not to scale). (**C**) Representative anti-H3K4me3 and H3K27me3 ChIP profiles (normalized to H3 ChIP) and Suz12 ChIP profiles (% Input; n = 3) for three independently transfected cell lines. Shaded box includes primers spanning the Artificial CGI. ChIP control amplicons are derived from the TSS of the active genes Sox2 (S) and GAPDH (G); the TSS of bivalent gene Hoxc8 (H) and an inconspicuous negative control region on mouse chromosome 15 (−). Error bars indicate the standard deviation of PCR replicates. (**D**) Bisulfite sequencing of the three cell lines shown in (**C**). In the map above, blue strokes show CpGs in the CGI-like insert and the clear box indicates the bisulfite amplicon. Methylated and unmethylated CpGs are depicted as filled and open circles, respectively.

The following source data and figure supplements are available for figure 1:

**Source data 1**.

**Figure supplement 1**. Bivalent chromatin at artificial CGI-like sequences in mouse ESCs.

**Figure supplement 2**. Synthetic DNA elements with different sequence properties.

---

replicate cell lines. None displayed a peak over the insert, whereas control active genes were RNA polymerase-positive as expected (*Figure 2A* and *Figure 2—figure supplement 1*). Consistent with the absence of RNA polymerase, we did not detect significant peaks of histone acetylation over the CGI-like insertions (*Figure 2B*). In summary, data derived from three distinct promoter-less CGI-like sequences indicate that bivalent chromatin is the default state for transcriptionally inert CGI-like DNA in ESCs.

We next asked whether the CXXC protein Cfp1 is enriched at the CGI-like sequences. To facilitate detection of Cfp1, we introduced the BAC containing the artificial CGI into a transgenic cell line expressing a Cfp1-GFP fusion protein (*Denissov et al., 2014*). Having verified that a bivalent domain was formed at the artificial CGI in these cells (data not shown), ChIP was performed on three independent cell lines using an anti-GFP antibody. We consistently observed discrete enrichment of Cfp1 at the CGI-like insertion (*Figure 2C*). To determine whether the formation of a bivalent domain at the inserted artificial CGI-like sequence was dependent on Cfp1, the artificial CGI was introduced into *Cfp1−/−* mouse ES cells (*Carlone and Skalnik, 2001*) and three independent lines were analysed by ChIP. H3K4me3 levels at the artificial CGI were clearly detectable in the *Cfp1−/−* cells, indicating that Cfp1 is not required for the formation of H3K4me3 levels at the bivalent domain (*Figure 2D*). Depletion of Cfp1 was previously reported to preferentially cause a decrease of H3K4me3 at active genes without affecting non-productive genes (*Clouaire et al., 2012*). In agreement with this finding, we observed reduced H3K4me3 at the active control gene *Sox2* compared to wildtype cells (Compare *Figures 2D and 1C*). We also followed the fate of chromatin modifications during differentiation of ESCs to neuronal progenitor cells (*Figure 2—figure supplement 1B*) and found a consistent drop in H3K4me3 accompanied by persistent or increased H3K27me3 (*Figure 2—figure supplement 1*). This transition from bivalency to H3K27me3 marking alone matches that at native CGI-associated genes that remain transcriptionally silent during differentiation (*Bernstein et al., 2006*).

## A high CpG frequency is necessary for the creation of bivalent domain

Although G + C content and CpG frequency are related features, they can be varied independently (*Figure 1A*). To establish the importance of these features for determination of bivalent chromatin, we varied CpG frequency and G + C content in 1000 base pair long artificial DNA sequences (*Figure 1—figure supplement 2* and *Figure 1—source data 1*). An artificial CGI with a base composition similar to that of a normal CGI (65% G + C) but with a low density of CpGs, similar to that of the bulk genome (1 CpG/100 bp), was designed (Low CpG / High G + C). This Low CpG / High G + C sequence failed to create bivalent chromatin as neither H3K4me3 nor H3K27me3 was detected in three independent ESC lines (*Figure 3A*). We note that the relative values of control and experimental data points are consistent between experiments although we observe variability in the absolute precipitation levels due to the use of different antibody suppliers between experiments over an extended time period. Our conclusion from this data is that a G + C-rich base composition alone is insufficient to recruit either H3K4me3 or H3K27me3.

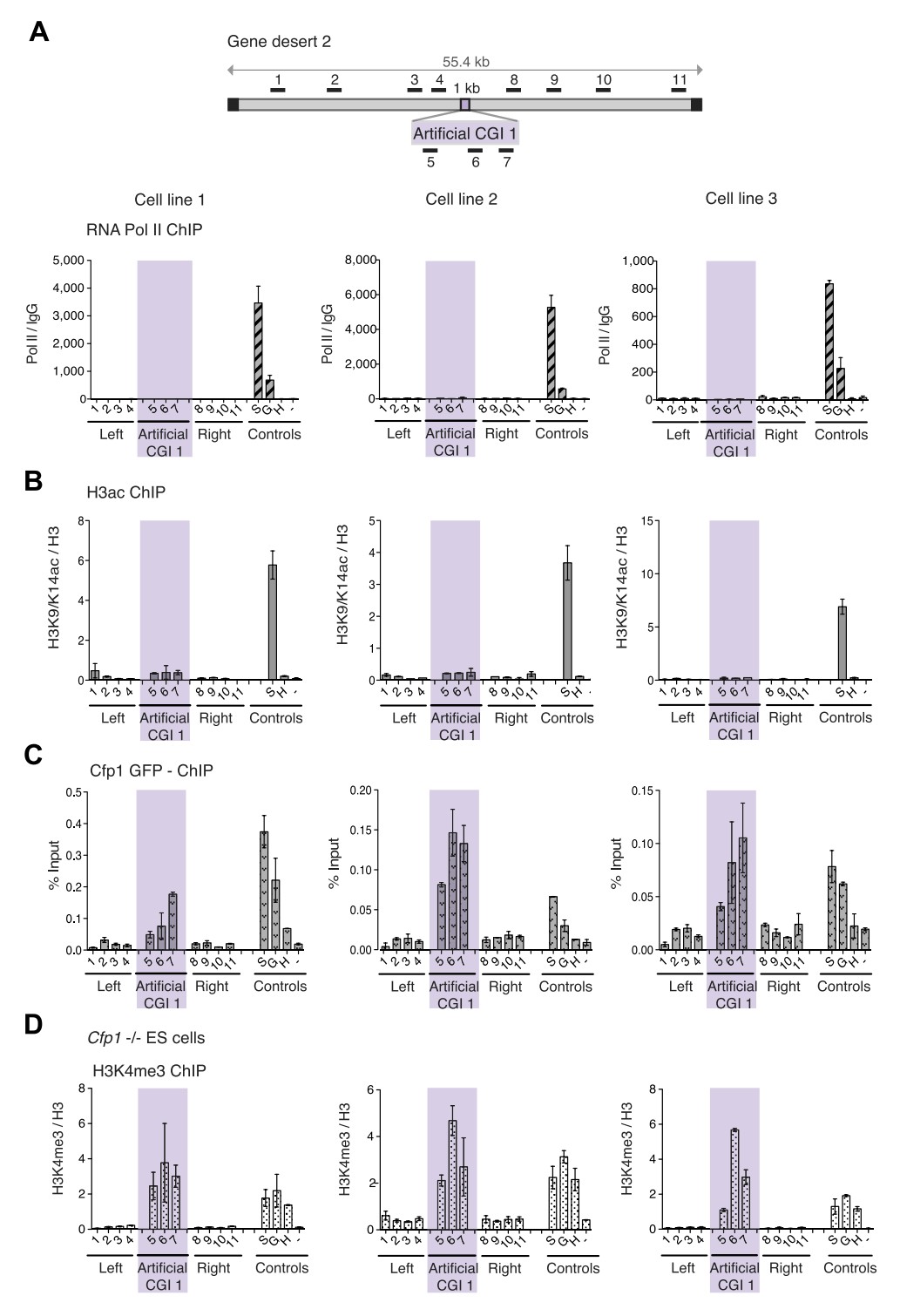

**Figure 2**. H3K4me3 at a promoter-less artificial CGI forms independently of *Cfp1* and RNA polymerase II. (**A**) Map of gene desert 2 with integrated Artificial CGI-like construct labeled as in *Figure 1B*. Representative ChIP with an antibody specific for the N-terminus of RNA polymerase II for three independent cell lines (% Input over IgG; n = 2). (**B**) Representative anti-H3K9/K14 acetylation ChIP profiles for three independently transfected cell lines normalized to H3 ChIP (n = 2). (**C**) Mouse ES cells expressing GFP-tagged *Cfp1* were transfected with Artificial CGI construct and bound Cpf1 was assayed by ChIP with anti-GFP antibodies in three independent cell lines
*Figure 2. Continued on next page*

*Figure 2. Continued*

(n = 2). (**D**) Representative anti-H3K3me3 ChIP for three independent *Cfp1−/−* mouse ES cells transfected with the Artificial CGI construct (n = 2). Control ChIP amplicons are as in *Figure 1C*. Error bars indicate standard deviation of PCR replicates.

The following figure supplement is available for figure 2:

**Figure supplement 1**. H3K4me3 at an artificial CGI independent of *Cfp1* and RNA polymerase II.

## A + T-rich CGIs become reproducibly DNA methylated

This result raised the possibility that CpG frequency alone determines the chromatin state, with G + C content playing no role. To test this idea, we generated four different artificial DNA sequences that were CpG-rich to the same level as typical CGIs (10 CpGs/100 bp), but relatively A + T-rich in overall base composition (three of 40% and one of 50% G + C on average; *Figure 1—figure supplement 2* and *Figure 1—source data 1*). Contrary to expectation, none of these insertions generated a focus of bivalent chromatin in multiple independent cell lines (*Figure 4A* and *Figure 4—figure supplement 1*). A potential explanation for this finding came from an analysis of DNA methylation status, which showed that in replicate cell lines the CGIs had all become densely methylated at CpGs (*Figure 4B* and *Figure 4—figure supplement 2*). The striking contrast between the consistent methylation-free status of three separate G + C-rich, CpG-rich integrants and the reproducible dense methylation of four unrelated A + T-rich, CpG-rich sequences of the same length indicates that base composition is a strong determinant of DNA methylation status. A plot of G + C-content against percentage CpG methylation showed a sharp transition between 50 and 60% G + C (*Figure 4D*). Interestingly, a CGI-like insertion with a base composition of 55% G + C (MeCP2-eGFP) studied previously (*Thomson et al., 2010*) showed an intermediate DNA methylation level, suggesting that it lies on the transition point for triggering de novo methylation (*Figure 4D*).

## Removal of DNA methylation restores the bivalent domain

The results indicate that DNA methylation is dominant over both H3K4me3 and H3K27me3, as in its presence neither chromatin mark is observed at the artificial CGIs. To test this hypothesis, we asked if removal of DNA methylation could restore bivalent chromatin at A + T-rich CpG-rich sequences by using mutant ESCs lacking the de novo DNA methyltransferases Dnmt 3a and Dnmt 3b (*Okano et al., 1999*), which display severe DNA hypomethylation (*Figure 4—figure supplement 2*). To ensure that histone modifying enzyme activities are not disrupted in the *Dnmt 3a/3b* double knock out cells, we generated stable cell lines with the artificial CGI-containing BAC and confirmed that a bivalent domain was observed over the insertion (*Figure 4—figure supplement 2*). When the High CpG /High A + T construct was introduced into Dnmt 3a/3b knock out cells it now remained unmethylated, and importantly, bivalent chromatin was detected at the inserted sequence (*Figure 4C*). We noticed that H3K4me3 was reproducibly weaker than controls in *Dnmt 3a/Dnmt 3b* double mutant cells, whereas a robust H3K27me3 signal was obtained. This may indicate that A + T-rich DNA is less able to recruit H3K4me3 despite its high CpG density. The dominance of the polycomb-associated H3K27me3 mark at A + T-rich CGIs was also seen when DNA methylation was partially reduced by growing cells for 10 days in 2i medium, which enhances the pluripotent state. In line with previous reports (*Ficz et al., 2013*; *Habibi et al., 2013*), we found that the average level of genomic CpG methylation was reduced from ~95% to ~55% (*Figure 4—figure supplement 2*). Whereas cells grown in serum-containing medium lacked both the tested histone marks, 2i-grown cells displayed a strong increase in H3K27me3 without the appearance of noticeable H3K4me3 (*Figure 4—figure supplement 2*). We conclude that while CpG frequency is a key feature of CGIs that determines the signature chromatin marks at CGIs, A + T-richness confers an intrinsic susceptibility to de novo methylation that is dominant over these chromatin modifications (*Figure 4E*).

## Status of endogenous DNA domains with atypical CpG or G + C composition

Our study assessed the role of G + C-richness vs CpG-richness on chromatin structure by experimentally varying these sequence parameters individually in synthetic DNA domains. The question arises:

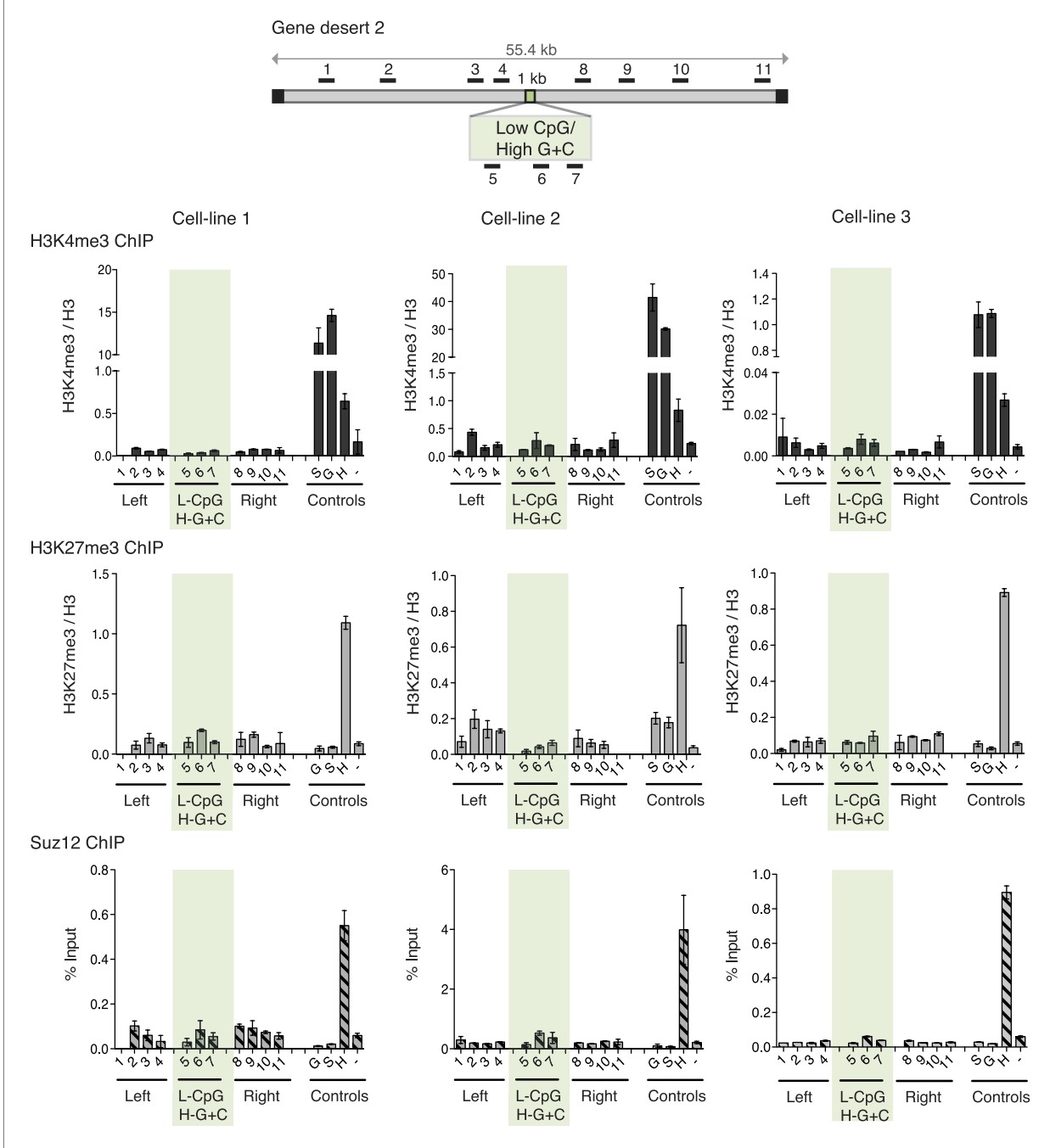

**Figure 3**. High G + C content is not sufficient to create a bivalent chromatin domain. Map of gene desert 2 showing the integration site of the Low CpG / High G + C (L-CpG / H-G + C) construct labeled as in *Figure 1B*. Representative anti-H3K3me3 and H3K27me3 ChIP profiles (normalized to H3) and Suz12 ChIP profiles (% Input; n = 3) are shown for three independent transfected cell lines. Shaded bar includes primers spanning the Low CpG / High G + C construct. Control ChIP amplicons are as in *Figure 1C*. Error bars indicate standard deviation of PCR replicates.

do equivalent atypical sequences exist naturally in the mouse genome and if so what is their chromatin modification status? We searched first for G + C-rich (≥61%), CpG deficient (≤1/100 bp) sequences of ≥500 bp in length and identified 1954 examples. None of these coincided with bivalent chromatin in ESCs, in agreement with our results using synthetic DNA. To determine if A + T-rich, CpG-rich sequences also exist naturally in mice, we divided the genome into windows of 100 bp and calculated the G + C content and CpG density of each. To ensure that we selected regions with profiles

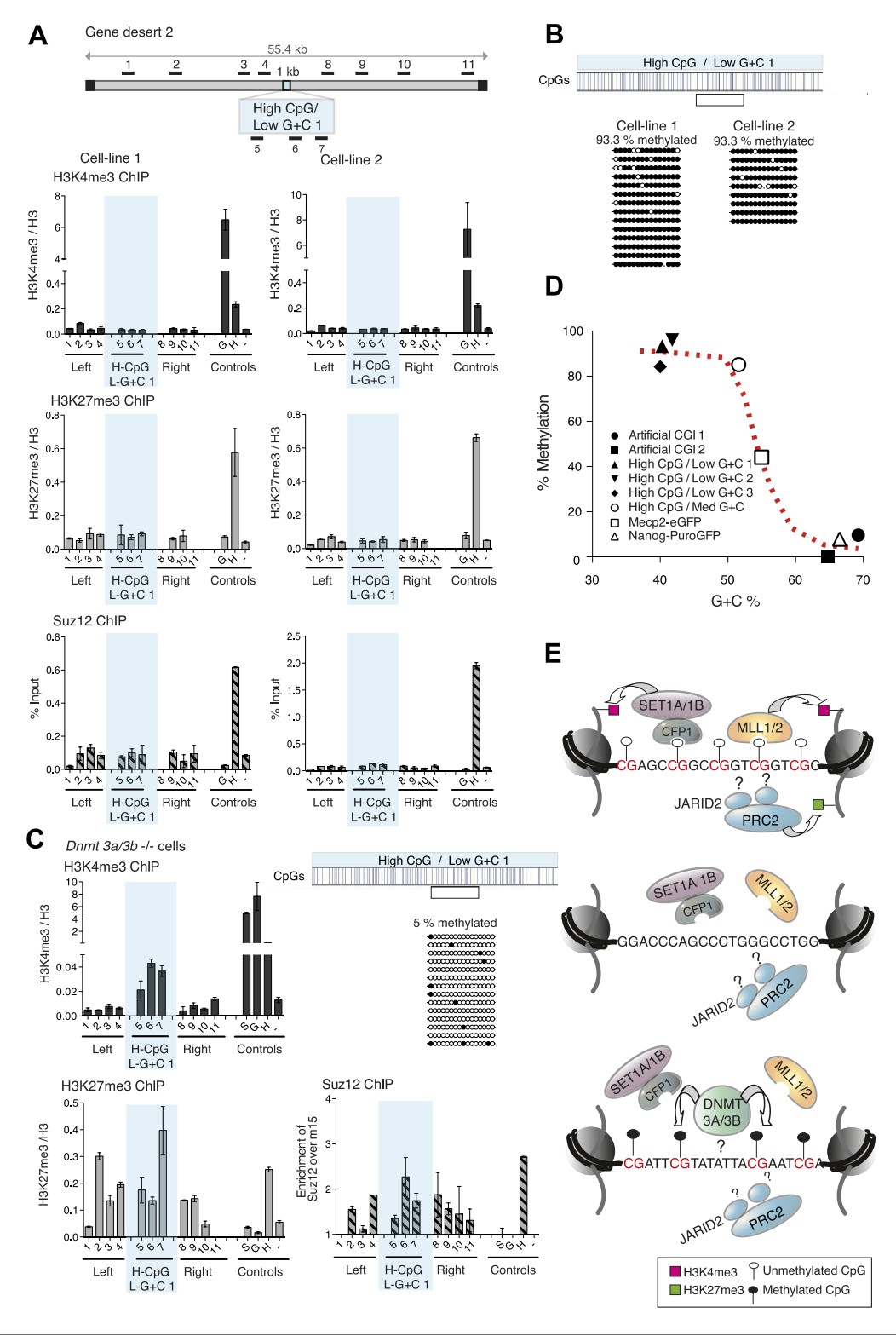

**Figure 4**. CpG-rich DNA sequences on an A + T-rich background fail to form bivalent chromatin and reproducibly acquire DNA methylation. (**A**) Above: Map of gene desert 2 indicating the integration site of the High CpG / Low G + C 1 (H-CpG / L-G + C 1) construct as in *Figure 1B*. Representative anti-H3K4me3, H3K27me3 and Suz12 ChIPs shown (n = 3) for each of two independently transfected cell lines (Third line not shown). The shaded bar includes

*Figure 4. Continued on next page*

*Figure 4. Continued*

primers spanning the High CpG / Low G + C 1 construct. (**B**) Bisulfite sequence analysis of the two cell lines shown in (**A**). Clear box indicates bisulfite amplicon. In the map above, blue strokes show CpGs in the CGI-like insert and the clear box indicates the bisulfite amplicon. Methylated and unmethylated CpGs are depicted as filled and open circles, respectively. (**C**) The High CpG / Low G + C construct was integrated into *Dnmt 3a–/– Dnmt 3b–/–* double mutant mouse ES cells. Representative H3K4me3, H3K27me3 and Suz12 ChIPs are shown (n = 3). Upper right panel shows bisulfite sequence analysis of a cell line containing the High CpG / Low G + C construct in *Dnmt 3a/b –/–* cells, presented as in panel (**B**). (**D**) The relationship between G + C content of constructs analysed in this study and their DNA methylation status. Data for Mecp2-eGFP and Nanog-PuroGFP refer to cell lines reported previously (*Thomson et al., 2010*), but reanalyzed for this study. (**E**) Diagrams depicting the influence of CGI sequence composition on chromatin structure. Upper panel: Sequences with high CpG frequency and high C + G content attract both H3K4 and H3K27 methyltransferase to establish bivalent chromatin domains and they remain unmethylated. SET1A/1B and MLL1/2 complexes contain CXXC domains that may target H3K4me3 to CGIs. The mechanism by which the PRC2 complex is targeted is unknown. Middle panel: Without CpGs, H3K4 and K27 methyltransferases are not recruited even when the DNA is G + C-rich. Lower panel: A + T-rich DNA fails to form a bivalent chromatin structure, even when the CpG density is high and is consistently subject to de novo methylation.

The following figure supplements are available for figure 4:

**Figure supplement 1**. CpG-rich, A + T-rich DNA sequences do not form bivalent chromatin.s.

**Figure supplement 2**. DNA methylation at CpG-rich, A + T-rich DNA sequences blocks bivalent chromatin.

**Figure supplement 3**. CpG density and CGI length at bivalent CGIs correlate positively with H3K4me3 and H3K27me3 levels in mouse ESCs.

consistently above the thresholds (as is the case for our artificial constructs) rather than short CpG dense regions flanked by AT rich DNA, we subtracted windows with ≥50% GC content or <5 CpGs and searched for blocks of adjoining windows. Uniformly A + T-rich and CpG-rich DNA sequences of this kind were absent in the mouse genome. Even when the criteria were relaxed to include sequences of only 500 base pairs in length (the approximate minimum size of CGIs), we found no examples. As A + T-rich regions are consistently DNA methylated, it seems likely that, in the absence of strong selection, the high mutation rate of 5-methylcytosine effectively resists CpG accumulation at these domains over evolutionary time.

## Discussion

Epigenetic marks are often considered to be sensitive to both developmental and environmental signals. Our data reinforce the complementary view that the underlying genetic information, independent of nurture or developmental history, plays an important role in setting up CGI chromatin structures. In ESCs at least, each of the major CGI chromatin configurations is informed by DNA sequence. We find that the non-methylated state of CGIs depends on G + C-rich base composition and that both the signature histone marks H3K4me3 and H3K27me3 depend on a high density of the non-methylated CpG dyad. The importance of primary DNA sequence in determining bulk features of the epigenome have been suggested elsewhere. For example, DNA sequence motifs that are recognised by transcription factors strongly influence DNA methylation patterns (*Lienert et al., 2011*; *Stadler et al., 2011*) and inter-individual variation in other epigenomic features maps to sites of human DNA sequence heterogeneity that are probably causal (*Kasowski et al., 2013*; *Kilpinen et al., 2013*; *McVicker et al., 2013*). Our data show that even gross features of genomic DNA, such as base composition and the frequency of the 2-base pair sequence CpG, influence the epigenome.

Do our findings reflect the relationship between naturally occurring DNA sequences and chromatin modification? Almost all native CGIs in mouse ESCs are non-methylated at the DNA level and coincide with peaks of H3K4me3, which is often seen as the signature histone mark of CGIs (*Bernstein et al., 2006*; *Thomson et al., 2010*). About one third of the H3K4me3-marked CGIs in mouse ES cells also carry H3K27me3 and are therefore defined as bivalent (*Ku et al., 2008*). H3K27me3 is usually relatively dispersed compared with the discrete localisation of components of the PRC2 complex, which deposit this mark. It is estimated that at least 97% of peaks corresponding to the PRC2 component Ezh2

coincide with CGIs (*Ku et al., 2008*). This has led to the suggestion that polycomb is targeted to G + C-rich or CpG-rich DNA (*Tanay et al., 2007*; *Mendenhall et al., 2010*; *Lynch et al., 2012*; *Long et al., 2013*). It was shown previously that CpG density within the non-methylated CGI fraction as a whole is proportional to the H3K4me3 ChIP-seq signal in both mouse and human cells (*Illingworth et al., 2010*). We asked whether the same relationship holds true for bivalent CGIs and whether it also applies to H3K27me3. Within a set of 2547 exclusively bivalent promoters from mouse ESCs (*Marks et al., 2012*; *Denissov et al., 2014*), we found that 92% coincide with CGIs. Within this bivalent group, CGI length and CpG density both correlate positively with H3K4me3 and also H3K27me3 levels (*Figure 4—figure supplement 3*). In contrast to typical CGI-like sequences, endogenous G + C-rich, CpG-poor sequences do not display a bivalent chromatin structure. A + T-rich, CpG-rich domains of CGI-like dimensions, however, are effectively absent from the mouse genome. The sum of available data argues strongly that a high abundance of the dinucleotide CpG is a key precondition for the formation of bivalent chromatin.

A possible mechanism for recruitment of H3K4me3 involves DNA binding by H3K4 methyltransferases, each of which is associated with a CpG-binding CXXC domain. In Mll1 and Mll2 the DNA binding domains are within the SET-containing protein themselves (*Allen et al., 2006*; *Cierpicki et al., 2010*), whereas in the case of Set1/COMPASS the CXXC domain resides in the Cfp1 protein component of the multi-subunit complex (*Lee and Skalnik, 2005*). The H3K4me3 component of bivalent CGIs in ESCs depends on Mll2 (*Hu et al., 2013*; *Denissov et al., 2014*). Accordingly, we have found that bivalent CGIs form normally in *Cfp1−/−* ES cells. Since the binding of the CXXC domain to CpG is abolished by methylation of the cytosine moiety (*Lee et al., 2001*), this may explain why DNA methylation prevents the formation of H3K4me3 even in the presence of high CpG densities. The mechanism responsible for recruiting PRC2, which is the complex responsible for the establishment of H3K27me3, remains uncertain, as no CpG binding component of PRC2 has yet been detected. (*Thomson et al., 2010*; *Hu et al., 2013*; *Wu et al., 2013*). A recent study, however, indicates that the CXXC-domain protein KDM2B can target the PRC1 complex to CGIs and recruit PRC2 secondarily, which would accord with our findings (*Blackledge et al., 2014*).

Bivalent chromatin has attracted attention due to the proposal that it represents a poised transcriptional state in pluripotent cells (*Bernstein et al., 2006*; *Voigt et al., 2013*). It is suggested that the poised state is resolved during differentiation as the affected gene becomes either transcriptionally activated or repressed. Recent evidence has clearly established that H3K27me and H3K4me can be present on the same nucleosome, albeit on different histone tails (*Voigt et al., 2012*). The biological significance of bivalent chromatin has recently become less certain, however. Whereas silent CGI promoters in ESCs grown in serum usually exhibit a bivalent chromatin structure, this chromatin configuration is significantly reduced in 2i medium, which discourages differentiation and is thought to induce a more pronounced pluripotent state (*Marks et al., 2012*). Also, cells in which the histone methyltransferase Mll2 is depleted lose H3K4me3 at hitherto bivalent CGIs, but this has no detectable effect on the induction kinetics of the associated genes upon differentiation (*Hu et al., 2013*). The evidence remains inconclusive, but it raises the possibility that bivalency is not an essential precondition for gene activation during differentiation. Here we find that bivalency is not confined to poised developmental genes but is a default response to any CpG-rich DNA sequences, even when these are completely artificial. It is apparent from this that the bivalent chromatin structure is not reserved for developmentally important genes, but is a response to general features of local DNA sequence.

Bulk mammalian DNA has a base composition of 40% G + C and is globally methylated at CpGs to an average level of ~65%, whereas G + C-rich CGIs are usually DNA methylation-free. The transfection experiments reported here recapitulate this distinction as G + C-rich CGI-like DNA reproducibly resisted DNA methylation, whereas A + T-rich DNA reliably became densely methylated. Our findings raise the possibility that this broadly binary pattern of global DNA methylation may be determined by base composition. CpG density did not affect this susceptibility to methylation, which occurred even when CpG occurred at densities typical of endogenous CGIs. Two simple alternative explanations are possible: either A + T-rich DNA attracts de novo methylation, or G + C-richness excludes it, leaving the bulk genome to be methylated by default. Both Dnmt 3L and Dnmt 3A have the potential to be repelled by H3K4me3 (*Jia et al., 2007*; *Ooi et al., 2007*), suggesting that recruitment of H3K4 methyltransferases with CpG-binding CXXC domains protects CGIs against de novo methylation. We find, however, that A + T-rich CpG-rich sequences are reproducibly DNA methylated despite their ability to attract H3K4me3 when DNA methylation is absent. Since these sequences were initially

non-methylated when transfected into cells, it may be that H3K4me3 alone is not sufficient to exclude CpG methylation from these regions. This would accord with our previous observation that a moderately G + C-rich artificial CGI when inserted into ES cells was extensively methylated despite the presence of H3K4me3 on non-methylated copies (*Thomson et al., 2010*). Alternatively, A + T-rich CpG-rich sequences may attract H3K4me3 less robustly than G + C-rich CGIs, thereby allowing access to de novo DNA methyltransferases.

The CGI phenomenon is conserved throughout the vertebrate lineage, but interestingly the G + C-richness characteristic of mammalian and bird CGIs is not seen in some vertebrate groups (*Long et al., 2013*). Fish for example have non-methylated CGI-like sequences at promoters, but these are not markedly G + C-rich compared with the bulk genome (*Cross et al., 1991*; *Long et al., 2013*). Since A + T-rich sequences are susceptible to de novo DNA methylation in mammals, it follows that some vertebrate groups may rely on a different set of mechanisms to prevent methylation at G + C-poor CGIs. Identifying potential components that discriminate between mammalian CGIs and the bulk genome is a priority for future work.

## Materials and methods

### Mouse ES cell lines

ES cells were grown in gelatinized dishes in Glasgow MEM (Gibco, UK) supplemented with 15% fetal bovine serum (Hyclone; Fisher Scientific, UK), 1% sodium pyruvate, 1% non-essential amino acids, 0.1% β-mercaptoethanol, 100U/ml penicillin, 100 µg/ml streptomycin and leukemia inhibitory factor (LIF). E14TG2a ES cells were used as wild-type ES cells. *Cfp1*−/− cells were a gift from David Skalnik and have been described previously (*Carlone et al., 2005*). *Dnmt 3a/3b* double knock out mouse ESC (DKOs) were as described (*Okano et al., 1999*). TC1-mES cells (a gift from Dr Ann Dean) with the hygromycin/thymidine kinase cassette in the β-globin locus (*Lienert et al., 2011*) were used for recombination-mediated cassette exchange. *Cfp1*-GFP tagged mouse ES cells have been described (*Denissov et al., 2014*). This integrated BAC contains the whole *Cfp1* gene with regulatory elements and GFP fused to the last codon of to create a C-terminal GFP tag.

### Recombineering

Custom Perl scripts were used to create random sequences with specific frequencies of CpG, G + C content and length (https://github.com/swebb1/cpg_tools). *Supplementary file 1* lists the CGI-like sequences used in this study. Artificial DNAs were synthesised (GeneArt; Life Technologies, UK) and cloned into a plasmid containing a selection cassette and homology arms for recombineering. CGI-constructs were introduced into human gene desert BACs by recombineering using the Red/ET system by Gene Bridges (Germany). Briefly, bacteria containing the gene desert BAC of interest (gene desert 1 mChr1:81,106,616-81,153,886 or gene desert 2 mChr18:36,042,881-36,175,341) were cultivated in LB medium plus chloramphenicol at 37°C o/n. On the next day 40 ng Red/ET plasmid were electroporated (1350 V, 10 µF, 600 Ohms) into the cells containing the BAC and incubated in LB containing chloramphenicol and tetracycline at 30°C for at least 15hr. Next day 50 µl of 10% L-arabinose were added to induce the expression of the Red/ET recombination proteins and samples were incubated at 37°C for 1h. Cells were electroporated with 200–300 ng of linearized DNA containing the CGI-like sequence, a kanamycin selection cassette and homology arms. Cells were re-suspended in 1 ml of SOC medium and recovered for 1–2h at 37°C. Cells were plated on plates containing chloramphenicol and kanamycin. Plates were incubated at 37°C o/n. Colonies were screened by colony PCR and control digests for successful recombination. The linearized BAC containing the CGI-like constructs and a selection cassette was used for transfection of mESCs.

### ES cell transfection

Linearized BAC DNA (0.5–2 µg) containing the CGI-like sequences and a selection cassette flanked by Frt sites was used to transfect 60% confluent mouse ESC growing in a 6-well plate using Lipofectamine LTX Plus (Invitrogen, UK). DNA was made up with OptiMem (Gibco, UK) to 500 µl, 2.5 µl PLUS reagent were added and incubated for 5 min at RT. Afterwards 6.25 µl Lipofectamine were added, incubated for 30 min at RT and added to the ES cells. Cells were split in a range of different ratios (10–0.1% of transfected cells) 24hr after transfection and plated onto 10 cm$^2$ dishes. Next day, selection medium containing the appropriate antibiotic (G418 250 µg/ml or Blasticidin 3 µg/ml) was added and cells

were grown until colonies were ready to be picked. Clones were analysed for incorporation of the constructs by PCR and 2–3 independent clones with low copy number integration were selected for the excision of the selection cassette. Circular plasmid (50 µg) containing a eukaryotic expression cassette for Flp or Dre were added to $2 \times 10^7$ cells and electroporated at 250 V and 500 µF using a BioRad electroporator (GenePulser Xcell; Biorad, UK). Cells were left to recover for 20 min at RT and seeded at different dilutions. Next day 0.8 µg/ml puromycin were added for 48hr and cells were cultured until colonies were big enough for picking. Successful excision was confirmed by Southern blotting. Artificial CGIs for insertion into the beta-Globin locus via recombination mediated cassette exchange were synthesised (GeneArt) with flanking inverted loxp sites and cloned into pBSIISK+ and electroporated into TC-1 ES cells carrying a Hygromycin/Thymidine Kinase double selection cassette in the beta-Globin locus (*Lienert et al., 2011*). Cells ($4 \times 10^6$) were pre-selected with hygromycin for 10 days, electroporated with 25 µg L1-artificial CGI-1L construct and 20 µg pCAGGS-Cre and selection for positive clones with 3 µm Ganciclovir was started 2 days after electroporation and continued for 8–10 days. Clones were tested for successful insertion of artificial CGIs by PCR screen and Southern Blot. For differentiation of mESC into neural precursors cells were plated ($4 \times 106$ cells/dish) in 15 ml EB medium (ES cell medium with 10% FBS and no LIF). After 4 days in EB medium trans-retinoic acid (Sigma, UK) was added to start neuronal differentiation. Medium was changed every 2 days. On day 8 EBs were disrupted, trypsinized and used for formaldehyde crosslinking.

## Chromatin immunoprecipitation

Chemical crosslink of chromatin was performed for 10 min at room temperature by addition of formaldehyde to a final concentration of 1%. Crosslinking was stopped by addition of glycine to a final concentration of 0.125 mM. After 5 min incubation cells were washed twice with ice-cold 1× PBS. If acetylation was examined, sodium butyrate was added to the PBS. Cells were centrifuged for 5 min at 330×g at 4°C and washed once in wash buffer A (0.25% Triton X-100, 10 mM EDTA pH 8.0, 0.5 mM EGTA pH 7.5, 10 mM, HEPES pH 7.5) and B (0.2M NaCl, 1 mM EDTA pH 8.0, 0.5 mM EGTA pH 7.5, 10 mM HEPES pH 7.5). Each buffer contained PMSF to a final concentration of 1 µg/ml, 1× Proteinase Inhibitor Complete Mix (Roche, UK) and 10 mM sodium butyrate. Cells were re-suspended in lysis buffer (20% SDS, 0.5 M EDTA, 1 M Tris pH 8.1, 1× complete protease inhibitors, 1 µg/ml PMSF, 10 mM sodium butyrate). Chromatin was sheared to 400–600 base pair fragments by sonication of the cell lysate using a twin Bioruptor (Diagenode, Belgium) for 15 cycles, 30 s on, 30 s off on high setting. The lysate was centrifuged for 10 min at 16,000×g at 4°C to collect cellular debris. Chromatin concentration was measured on a Nanodrop spectrophotometer. Chromatin (30 µg or 100 µg) was used for ChIP with antibodies against histone modifications or other proteins. For the ChIP, chromatin was made up to 100 µl with lysis buffer and 900 µl of dilution buffer (20 mM Tris–HCl, pH 8, 150 mM NaCl, 1% Triton X-100, 1 mM EDTA) were added. Antibodies were added as specified and samples were incubated o/n at 4°C on a rotating wheel. The next day debris was removed by centrifugation for 5 min at 16,000×g and 50 µl dynabeads protein G (Life Technologies, UK) were added to the supernatant of the IPs and samples were incubated on a rotating wheel for 2 hr at 4°C. IPs were washed 3× with 1 ml ice-cold wash buffer 1 (20 mM Tris–HCl, pH 8, 150 mM NaCl, 1% Triton X-100, 1 mM EDTA, 0.1% SDS), 2× with ice-cold wash buffer 2 (20 mM Tris–HCl, pH 8, 500 mM NaCl, 1% Triton X-100, 1 mM EDTA, 0.1% SDS) and 1× with ice-cold TE. 100 µl or 50 µl of freshly prepared 10% Chelex 100 Resin (100–200 mesh, BioRad) were added to samples or input respectively. Samples were boiled for 12 min, cooled to RT. Proteinase K (2 µl of 20 mg/ml) was added and incubated at 55°C for 30 min while shaking. After heating to 100°C, samples were spun down and 60 µl supernatant were transferred to fresh tubes. Each sample was made up to 300 µl total volume with 10 mM Tris pH 8, 0.1 mM EDTA and assayed by q-PCR.

## qPCR

*Supplementary file 1* lists the qPCR primers used in this study. qPCR was used to assess enrichment of specific regions in ChIP samples and to determine the copy number of BAC DNA inserted into the mouse genome. qPCR reactions (10 µl) contained SYBR Green SensiMix (Bioline, UK), 250 nM primers and 3 µl ChIP DNA or 100 ng DNA for the determination of copy numbers. PCR was carried out using a Roche Lightcycler and cycling conditions were as follows; initial denaturation at 94°C for 10 min followed by 45 cycles of denaturation at 94°C for 10 s, primer annealing for 10 s and primer extension at 72°C for 15 s. Using the Roche Lightcycler software, SYBR Green fluorescence measurements were plotted

relative to cycle number and the 2nd derivative maximum method was used to determine the cycle threshold values (Ct) for each sample. Values for duplicate of triplicate ChIP samples were calculated as %-input and copy number of artificial CGIs was assessed relative to Sox2 qPCR signal.

## Antibodies

α-H3K4me3 (Abcam[UK]-8580), α-H3K4me1(Abcam-8895), α-H3K27me3 (Millipore[UK]-07-449), α-H3K9/K14ac (Abcam 12,179), α-H3 (Abcam 1791), α-SUZ12 (Abcam 12,073-100), α-RNA Pol II N20 (Santa-Cruz[UK] 899), α-RNA Pol II S5P (Abcam 5131), α-RNA Pol II unphosphorylated CTD (Abcam 817), α-GFP (Chromotec [Germany] GFP-TRAP-A gta-20), α-IgG (Invitrogen 10500C).

## Bisulfite genomic sequencing

Bisulfite conversion of genomic DNA was carried out using the EpiTect Bisulfite Kit from Qiagen (UK). The converted DNA was used for PCR amplification of regions of interest, *Supplementary file 2* lists the CGI-like sequences used in this study. PCR products were gel-purified and cloned using the Stratagene blunt end cloning kit. Positive clones were sent for sequencing.

## Acknowledgements

We thank Ann Dean, Dirk Schubeler and David Skalnik for sharing cell lines, Sukhdeep Singh for bio-informatic insights and Martha Koerner, Matt Lyst and Sabine Lagger for critical comments on the manuscript. The work was supported by Grants from the Wellcome Trust (WT091580, WT84637, WT092076). E.W. was funded by a Wellcome Trust 4 year PhD studentship (WT086659). T.Q. holds a Marie Curie fellowship.

## Additional information

### Funding

| Funder | Grant reference number | Author |
|---|---|---|
| Wellcome Trust | WT091580 | Elisabeth Wachter, Timo Quante, Cara Merusi, Aleksandra Arczewska, Shaun Webb, Adrian Bird |
| Marie Curie Fellowship | 627442 ATRUN | Timo Quante |
| Wellcome Trust | WT086659 | Elisabeth Wachter |
| Wellcome Trust | WT092076 | Elisabeth Wachter, Timo Quante, Cara Merusi, Aleksandra Arczewska, Shaun Webb, Adrian Bird |
| Wellcome Trust | WT84637 | Elisabeth Wachter, Timo Quante, Cara Merusi, Aleksandra Arczewska |

The funders had no role in study design, data collection and interpretation, or the decision to submit the work for publication.

### Author contributions

EW, TQ, Conception and design, Acquisition of data, Analysis and interpretation of data, Drafting or revising the article; CM, AA, Acquisition of data, Analysis and interpretation of data, Drafting or revising the article; FS, Analysis and interpretation of data, Drafting or revising the article, Contributed unpublished essential data or reagents; SW, AB, Conception and design, Analysis and interpretation of data, Drafting or revising the article

## Additional files

**Supplementary files**
• Supplementary file 1. CGI-like DNA sequences.

• Supplementary file 2. Primers.

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
