## [Decision Letter]

Thank you for sending your work entitled “Synthetic CpG islands reveal DNA sequence determinants of chromatin structure” for consideration at *eLife.* Your article has been favorably evaluated by Chris Ponting (Senior editor), Anne Ferguson-Smith (Reviewing editor), and 2 reviewers.

The Reviewing editor and the other reviewers discussed their comments before reaching a decision. The comments were very positive and we are, in principle, keen to consider a revised version of your manuscript. The Reviewing editor has assembled the following comments to help you prepare a revised submission:

1) We are concerned about the generalization of the findings to endogenous genomic CGIs. It would be interesting to mine publicly available ChIP-seq data in ES cells to correlate H3K4me3/H3K27me3 peaks with CpG/G+C richness to test how endogenous (lowCpG/HighG+C) and (HighCpG/A+Trich) sequences (if they exist) correlate with bivalent chromatin. This analysis could be extended to ChIPseq data for H3K27me3 produced in Dnmt TKO ES cells (Hagarman JA, PLOS One 2013; Brinkman AB, Genome Res 2012) to test if removal of DNA methylation restores bivalent chromatin at endogenous HighCpG/A+Trich sequences.

2) Figure 1: Although generally unmethylated, the artificial CGI1 shows a high basal level of methylation (7-12%), which could reflect either real methylation or incomplete bisulfite conversion. This should be clarified, either by improving and repeating the bisulfite conversion/PCR/cloning or by measuring methylation at an endogenous control CGIs.

3) In Figure 3 can the authors comment on the variability of ChIP enrichments (H3K4me3 ChIP and Suz12 ChIP) obtained in the 3 cell lines?

---

## [Author Response]

*1) We are concerned about the generalization of the findings to endogenous genomic CGIs. It would be interesting to mine publicly available ChIP-seq data in ES cells to correlate H3K4me3/H3K27me3 peaks with CpG/G+C richness to test how endogenous (lowCpG/HighG+C) and (HighCpG/A+Trich) sequences (if they exist) correlate with bivalent chromatin. This analysis could be extended to ChIPseq data for H3K27me3 produced in Dnmt TKO ES cells (Hagarman JA, PLOS One 2013; Brinkman AB, Genome Res 2012) to test if removal of DNA methylation restores bivalent chromatin at endogenous HighCpG/A+Trich sequences*.

We have added consideration of this point as one extra section in the Results and additionally a paragraph in the Discussion. We also added a new supplement to Figure 4 (Figure 4—figure supplement 3).

In the Results section we have now identified a set of G+C-rich/CpG-poor sequences and found that these never coincide with bivalent chromatin domains.

On the other hand uniformly A+T-rich, CpG rich sequences like those used in our experiments do not occur in the genome, even when we relaxed the criteria considerably. Therefore we cannot ask if DNA methylation deficiency at these regions leads to the appearance of bivalent chromatin. Overall we hope it is clear that our approach to understanding the parameters that mediate the effects of CGIs on chromatin structure required us to vary each parameter in a way that may not occur naturally. Importantly though, our assay was conducted in vivo in ESCs using stably integrated sequences. The setting for our analysis is therefore native, even though the sequences themselves are unusual.

In the Discussion paragraph we collate evidence that the great majority of bivalent regions in ESCs are CGIs, and we have performed a new analysis of published data to show that the levels of H3K4me3 and H3K27me3 in bivalent CGIs correlate positively with CGI length and CpG density (new Figure 4—figure supplement 3). The available data align well with our results and previous evidence that CpGs are required to recruit both these chromatin marks to bivalent domains.

*2)*
Figure 1*: Although generally unmethylated, the artificial CGI1 shows a high basal level of methylation (7-12%), which could reflect either real methylation or incomplete bisulfite conversion. This should be clarified, either by improving and repeating the bisulfite conversion/PCR/cloning or by measuring methylation at an endogenous control CGIs*.

We repeated the bisulfite sequencing for these cell lines using as an internal control a region of the *Dlx5* CpG island, which we had previously shown to have a low level of DNA methylation (<5%). The results closely reproduced the DNA methylation levels reported in this manuscript and they showed lower levels of methylation at the *Dlx5* CGI as expected (see data below). Therefore, the somewhat higher-than-expected level of methylation at these artificial CGIs is replicated and is not due to incomplete bisulfite conversion (conversion levels averaged 98%). We now point out in the Results section that methylation of these artificial CGI-like sequences in a gene desert setting is somewhat higher than at an endogenous CGI in the same cells. In our opinion 10% methylation nevertheless qualifies as hypomethylated compared with the genome-wide average of ∼70%. Therefore this modest level of DNA methylation does not affect the conclusion that these sequences are hypomethylated. We do not plan to add the new bisulfite data to the manuscript.Percentage of CpG methylation in ampliconsControl CGIArtificialCGI (New data)ArtificialCGI (previous data)Cell line 13.98.59.2Cell line 25.07.97.2Cell line 34.512.512.4

*3) In*
Figure 3
*can the authors comment on the variability of ChIP enrichments (H3K4me3 ChIP and Suz12 ChIP) obtained in the 3 cell lines?*

We acknowledge that in Figure 3 the proportions of input DNA that are immunoprecipitated in different cell lines vary over a wide range. The reason is that these studies were not carried out in parallel, but over almost 12 months, using chromatin samples prepared at different times and assayed with different antibodies (often from different suppliers) with varying affinities. In particular, deletion of the neomycin cassette in cell line 3 initially failed and there was a delay while this step was repeated during which we had to switch H3K4me3 antibodies. In our opinion the interpretation of the results is not compromised by the observed variability, which in our hands is inherent to ChIP experiments on samples of diverse origin with changing reagents. The important point with respect to our conclusions is that the relative values of control and experimental data points are consistent between experiments even when absolute precipitation levels vary. We now draw attention to the variability in the Results section and offer a brief explanation.